# Do You Feel Safe Here? The Role of Psychological Safety in the Relationship between Transformational Leadership and Turnover Intention Amid COVID-19 Pandemic

Abu Elnasr E. Sobaih [1,2,*], Hassane Gharbi [1,3] and Ahmed E. Abu Elnasr [4]

[1] Management Department, College of Business Administration, King Faisal University, Al-Ahsa 31982, Saudi Arabia; hgharbi@kfu.edu.sa or hassane.gharbi@yahoo.fr
[2] Hotel Management Department, Faculty of Tourism and Hotel Management, Helwan University, Cairo 12612, Egypt
[3] Management Department, School of Business, University of Sfax, Sfax 3018, Tunisia
[4] Higher Institute for Specific Studies, Future Academy, Cairo 11771, Egypt; dr.ahmed.abulnasr@fa-hiss.edu.eg
* Correspondence: asobaih@kfu.edu.sa or abuelnasr@hotmail.co.uk

**Abstract:** Despite their significant role in the performance of hotel industry, hotel workers are suffering from high rates of turnover, due to several reasons, particularly amid the COVID-19 pandemic. The pandemic has had numerous negative consequences on hotel workers, including their intention to leave the job or/and career. This study is an attempt to investigate the impact of transformational leadership on turnover intention amid COVID-19 and how psychological safety can intermediate this relationship. The study used a quantitative research approach via a pre-test instrument, self-distributed and collected from hotel workers at different regions in the Kingdom of Saudi Arabia (KSA). Valid responses from 1228 workers, analyzed through a structural equation modeling (SEM) of AMOS version 23, showed that transformational leadership has a significant negative impact on turnover intention as hypothesized. Nevertheless, it has a significant positive impact on psychological safety, whereas psychological safety has a significant negative impact on turnover intention. The most important finding of this study was the perfect mediating effect of psychological safety in the link between transformational leadership and workers' turnover intention. This finding confirms that psychological safety has the ability to change the above-mentioned link. In other words, the presence of psychological safety ensures negative turnover intention, even if transformational leadership practices do not exist. The findings have implications for scholars and practitioners, especially in tourism and hotel context, in relation to the role of psychological safety and transformational leadership in creating a sustainable working environment to maintain a lower turnover intention.

**Keywords:** psychological safety; transformational leadership; turnover intention; social exchange theory; hotel industry; Kingdom of Saudi Arabia (KSA)

## 1. Introduction

The worldwide pandemic of novel coronavirus disease (COVID-19) has been fast spread and has affected nearly all countries and business sectors. This long-lasting pandemic seriously hit the global economy; nonetheless, the hotel industry was one of the hardest hit industries (Sobaih et al. 2021). Consequently, many hotels were obliged to close down, while others have had mass workers' lay-offs (Sobaih et al. 2021). Thus, hotel managers have implemented alternative strategies to overcome the tough financial situation during and amid the pandemic period (Taylor et al. 2020). These strategies included reducing and cutting operational costs, firing part-time and temporary workers, decreasing salaries or laying off a considerable percentage of their workers (Hall et al. 2020). Past studies (e.g., Sverke and Hellgren 2002; Dubey et al. 2020; Alyahya et al. 2021; Aliedan et al.

2022) have shown that crises usually have a negative psychological impact on hotel workers and ultimately on their attitudes and behaviors. Nonetheless, surviving employees, who had the opportunity to stay in the organization during the crises became more stressed, anxious, experienced panic and negative emotions, did not feel safe in their place, and were working under extreme pressures both physically and psychologically (Greenberg et al. 2020; Howe et al. 2020). Recent research (Obeng et al. 2020; Tu et al. 2021; Alyahya et al. 2021; Aliedan et al. 2022) confirmed that, amid the COVID-19 pandemic, hotel employees became more stressed, felt insecure, worried about their continuity in the job, and thought about changing their career.

Previous research (see for example, Deery and Jago 2015; Gui et al. 2020; Sobaih et al. 2022) have shown that leaders with transformational leadership characteristics have a significant influence on worker attitudes and behaviors. Therefore, the relationship between transformational leadership with other organizational attributes have drawn the attention of researchers in last few decades. Earlier studies indicated a positive influence of transformational leadership on different job outcomes such as job satisfaction (Medley and Larochelle 1995); workers' motivation (Masi and Cooke 2000); organization effectiveness (Lowe et al. 1996); organizational learning (Zagoršek et al. 2009); psychological capital (Gom et al. 2021); and intention to stay (Sobaih et al. 2022). However, there is limited published research, to the best of the research team's knowledge, examining the relationship between transformational leadership and turnover intention amid COVID-19 on hotel workers through the effect of psychological safety. Although the COVID-19 pandemic has had severe psychological influences on hotel workers (Alyahya et al. 2021; Aliedan et al. 2022), the role of psychological safety has not been fully addressed, especially in the link between transformational leadership and turnover intention of hotel workers amid COVID-19. Recent studies (e.g., Hebles et al. 2022; Groh 2019; Winasis et al. 2020) have found a significant relationship between transformational leadership, turnover intentions, and psychological capital. Nonetheless, research on the interrelationship between these variables in hotel industry remains limited or even unavailable to the best of the researchers' knowledge. This study aims to fill in a knowledge gap about the role of psychological safety in the link between transformational leadership and turnover intention of hotel workers. This helps in understanding the antecedents of turnover intention among hotel workers in an attempt to properly manage the high cost of this turnover intention in hotel industry.

This research draws on Social Exchange Theory (SET) to test the direct impact of transformational leadership on turnover intention of hotel worker amid COVID-19 and indirect impact through psychological safety. The SET argues that an individual's attitude and behavior is a result of exchange processes between two parties (Blau 1968). Hence, leaders of organizations tend to maximize the benefits of this exchange with their workers and minimize the cost. According to Cropanzano and Mitchell (2005), the SET framework is one of the most continuing and widely used conceptual backgrounds to understand the role of leadership style, especially transformational leadership, in stimulating the attitudes and behaviors of workers as an exchange. Therefore, the majority of studies on organizational behavior, especially in relation to management practices and their effect on workers attitude and behaviors, have been examined through the lens of SET (Cropanzano et al. 2017). For example, SET was adopted to understand organizational citizenship behaviors (Organ 1988, 1990), justice (Tepper and Taylor 2003), leadership and organizational support (Ladd and Henry 2000), transformational/transactional leadership, and turnover intention (Sobaih et al. 2022).

Drawn on SET, this paper hypothesizes that transformational leadership negatively influences turnover intention of hotel workers, whereas the transformational leadership positively influences employees' psychological safety. Furthermore, the psychological safety has a negative impact on turnover intention. Additionally, it is also expected that psychological safety amid COVID-19 would have a mediating effect in the link between transformational leadership and turnover intention of hotel workers. The research contributes to the academic body of hospitality literature and the practices of the hospitality

industry in relation to the creation of an appropriate working environment in the new normal and minimizing the high rate of turnover among hotel workers. Thus, on the basis of the above discussion, the three research questions are as follows:

Research question 1: What is the impact of transformational leadership on turnover intention of hotel workers amid the COVID-19 pandemic?
Research question 2: What is the impact of psychological safety on turnover intention of hotel workers amid the COVID-19 pandemic?
Research question 3: How does psychological safety intermediate the link between transformational leadership and turnover intention amid the COVID-19 pandemic in the hotel industry?

To achieve the purpose of the research and answer the research questions mentioned above, the structure of the current research article will be as follows. The first part of the article introduces the research by highlighting the research problem and presenting its purpose as well as research questions. The second part presents the research theoretical framework by defining the research constructs and reviewing the relationship between the research variables. The third part presents the research method including data collection and analysis methods. The fourth part presents the findings of this study, including the final research structural model. The fifth part discusses the findings of the study and presents the theoretical and managerial implications. The sixth part concludes the study, gives the final remarks, presents the limitations and proposes directions for future research.

## 2. Review of Literature

### 2.1. Defining the Study Constructs

According to Podsakoff and MacKenzie (1997), transformational leadership is a process that enables a leader to stimulate individuals, groups, teams, and organizations significantly. In this regard, Burns (1978) defined transformational leaders as people who inspire and enable followers to achieve higher levels of performance. In order to be more effective in pursuing organizational goals, followers of transformational leaders are inspired to look beyond their own interests. There are different types of leadership while compared to other major leadership theories s (e.g., transactional leadership or laissez-faire leadership), transformational leadership is identified as the most effective type of leadership (Den Hartog et al. 1997). Leadership is regarded as one of the most important factors in directing an organizational performance by establishing a clear vision for the organization's short and long-term commercial operations (Bass and Riggio 2006). Because of the broad range of its effects on workers' behavior, transformational leadership has received the most attention among leadership theories and approaches (Kelloway et al. 2012).

Five decades ago, the concept of psychological safety was introduced by Schein and Bennis (1965), and from the beginning of the 1990s until the present it has seen renewed interest among psychological scholars (Edmondson and Lei 2014). In the context of the workplace psychological safety refers to the "*individual's self-belief about the workplace that it is safe to take the interpersonal risk, speak up the ideas, share opinions and act independently on key decisions*" according to Edmondson (1999). Consequently, creating a psychologically safe work environment is essential for lowering actual and perceived interpersonal risks (Grant and Ashford 2008).

Turnover intention is defined as "*the conscious and deliberate willfulness to leave the organization*" (Tett and Meyer 1993, p. 262). Furthermore, the subjective assessment of a person's likelihood of leaving the company in the near future is referred to as turnover intention (Mobley 1982). Hellman (1997) highlighted the issue of behavioral turnover intention that reveals a person's intention to leave the company. Moreover, the final cognitive stage of a voluntary turnover decision-making process is turnover intention (Griffeth et al. 2000).

*2.2. Transformational Leadership and Turnover Intention*

Employee turnover is a common problem for the hospitality industry worldwide (Haldorai et al. 2019) as turnover in the hospitality industry is almost double the average rate for other industries. Based on Ajzen (1980) theory, turnover intention referred to an employee's behavioral intention to leave an organization. Workers have influence over their turnover intentions, and behavioral intention is the main predictor of the behavior, in this instance the actual turnover. No doubt that high level of turnover will lead to low level of job satisfaction and worker moral, high personnel cost, which negatively impact on guest satisfaction (Yang et al. 2012). According to Khan (2015), little research has been conducted to address how transformational leadership predicts follower's turnover intent. On the other side, earlier studies in hospitality industry (e.g., Chen and Wu 2017; Zou et al. 2015; Tang et al. 2015) showed that transformational leadership has the potential to affect a worker's intention to leave. This could be manifested through transformative leadership qualities in idealized influence, inspirational motivation, intellectual stimulation, and individualized consideration. Being able to commend and attract employees through their charismatic characteristics, employees would want to identify with and emulate their role model (Chen and Wu 2017). Reduced job stress, reduced absenteeism, improved employee's operational efficiency, and increased employee self-esteem are typical outcomes of transformative leaders who care about their team members' wellbeing (DeGroot et al. 2000; Breevaart et al. 2014). Thus, the consideration offered by transformational leaders creates a solid foundation for the management and their employees to reduce employee intention to leave. Additionally, recent studies on hotel workers amid COVID-19 (Alyahya et al. 2021; Aliedan et al. 2022) showed that workers were negatively affected by inappropriate practices of their management and hence they have an intention to leave the job. Research on transformational leadership in the hotel context pre COVID-19 (Sobaih et al. 2022) showed that transformational leadership significantly affects worker intention to stay. Based on these arguments, we could propose that:

**Hypothesis 1 (H1).** *Transformational leadership practices negatively impact on turnover intention of hotel workers.*

*2.3. Transformational Leadership and Psychological Safety*

Research studies (e.g., Jyoti and Dev 2014; Men 2014; Moyo 2019) have described transformational leadership as a kind of leadership that organizes relationships in order to face outside and inside changes by inspiring, enhancing, and transforming the actions and aspirations of the employees by integrating empathy, compassion, and sensitivity, and building relationships through innovation. Thus, the leader and their workers have a shared vision and values, mutual respect, and trust. Leaders adopt transformational leadership using four dimensions: (1) inspirational motivation, (2) intellectual stimulation, (3) idealized influence, and (4) individualized consideration (Breevaart et al. 2013; Hay 2006; Jyoti and Dev 2014). Additionally, Kahn (1990) suggested that leadership style could be a predictor of a worker's psychological safety when taking into account the crucial role that psychological safety plays at work. Specifically, transformational leadership practices convey to staff members their value, which is an essential source of psychological safety (Yin et al. 2019). Workers are encouraged to express themselves freely in a secure work environment because transformational leaders tend to stimulate intellectual curiosity and challenge preconceptions (Zhang et al. 2011; Carmeli et al. 2014). Additionally, individual consideration stands for the idea that each person's unique needs must be taken into account (Avolio and Bass 1995). This means that transformational leaders value two-way communication (Bass and Riggio 2006). That is, when leaders exhibit individualized consideration and stimulate behaviors, workers will perceive high levels of psychological safety (Detert and Burris 2007). Workers can enhance their psychological safety under leaders who encourage, are coach-oriented, and are open to questions and challenges (Edmondson 1999). To that end, it is suggested that transformational leadership as a

contextual resource improves workplace environment and influence on psychological safety as a personal psychological resource (Edmondson 1999). Hence, workers may perceive a safe work environment if transformational leadership practices such as individualized consideration and intellectual stimulation are used. Workers should be encouraged to participate actively in their work because of this positive emotional experience related to psychological safety. Hence, the following could be proposed:

**Hypothesis 2 (H2).** *Transformational leadership practices positively impact on psychological safety of hotel workers.*

*2.4. Psychological Safety and Turnover Intention*

According to previous studies (e.g., Edmondson and Lei 2014) on psychological safety in organizational settings, workers need to feel safe to be able to grow, learn, contribute, and perform well in a rapidly changing world. In other words, they need to feel safe at their workplace. Furthermore, psychological safety has been linked to a variety of advantageous organizational outcomes, including promoting knowledge sharing within an organization as well as favorable effects on organizational performance (Baer and Frese 2003). Hence, psychological safety could be translated into a mechanism that lessens stress among employees, thereby lowering their dissatisfaction with their jobs and, consequently, their turnover intentions (Edmondson et al. 2001; Frazier et al. 2017). Psychological safety gives workers the assurance that they will be treated fairly, will not be humiliated or penalized for making decisions, can offer suggestions, or develop ideas for workplace-related problems. However, their actions will be supported and valued. Hence, the level of respect and trust between the leadership and followers are the results of this appreciation. On the other side, given the extensive body of research on the benefits of psychological safety at workplace, it is surprising that this factor was not included as a predictor in the thorough meta-analysis of turnover provided by Rubenstein et al. (2018). However, Hebles et al. (2022) reported that psychological safety may result in a mechanism that lessens stress among employees, thereby lowering their dissatisfaction with their jobs and consequently their intentions to leave. Furthermore, amid COVID-19 hospitality employees became more unsecured, stressed, and anxious (Aliedan et al. 2022). Consequently, greater stress increases employee's turnover intention, but this kind of employee feeling might be diminished through using different management tools to reduce employees' turnover intentions, such as encouraging workers to express their difficulties and struggles, i.e., a psychologically safe environment (Hebles et al. 2022). Therefore, it could be proposed that:

**Hypothesis 3 (H3).** *Psychological safety negatively impacts on turnover intention of hotel workers.*

*2.5. The Mediating Effect of Psychological Safety in the Link between Transformational Leadership and Turnover Intention*

Kahn (1990) defined psychological safety as "*feeling able to show and employ one's self without fear of negative consequences to self-image, status or career*" (p. 708). This description emphasizes the significance of perceiving reduced interpersonal risk because it primarily focuses on individuals' perceptions (Frazier et al. 2017). As a beneficial personal resource, psychological safety is a key explanation for how contextual factors influence different work-related outcomes (Ten Brummelhuis and Bakker 2012). Furthermore, transformational leaders convey to their staff members organizational value, which is an essential source of psychological safety (Yin et al. 2019). According to Frazier et al. (2017), it makes sense that supervisory support is a significant predictor of psychological safety. On the other side, several research studies have shown that transformational leadership has a direct relationship with workers' intention to turnover (Chen and Wu 2017; Zou et al. 2015; Tang et al. 2015; Khan 2015). The current paper can be considered the first attempt to investigate the indirect effect of transformational leadership on turnover intention through psychological safety. In this study psychological safety is expected to have a mediation role

in the link between transformational leadership and turnover intention. Hence, it could be hypothesized that:

**Hypothesis 4 (H4).** *Psychological safety has a mediating effect in the link between transformational leadership and turnover intention of hotel workers.*

The research theoretical model, which summarizes the research framework and shows the research hypotheses is shown in Figure 1.

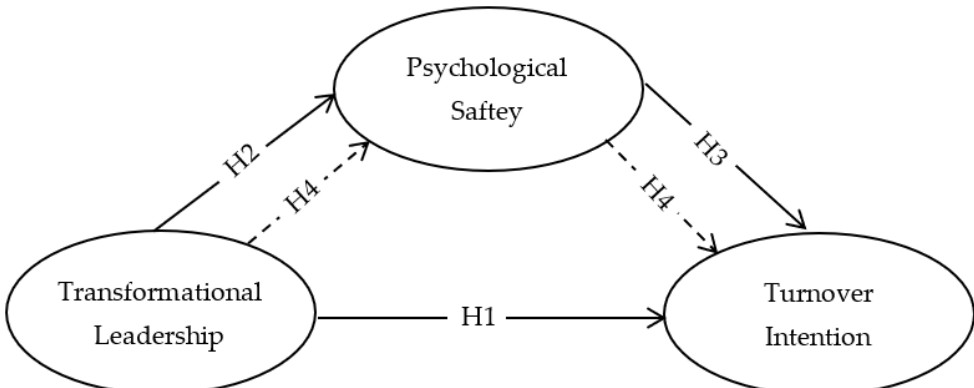

**Figure 1.** The theoretical model (developed based on the literature review; dotted lines reflect indirect effect).

## 3. Methodology

### 3.1. Sample and Data Collection

As discussed earlier, the current study is concerned with hotel workers. Since the research team is located in the Kingdom of Saudi Arabia (KSA), workers in large hotels were the population of the current research for pragmatic reasons. It was argued that large hotels are more likely to adopt proper management practices than small hotels, which include the adoption of transformational leadership practices (Sobaih 2018; Sobaih et al. 2022). The current study is concerned with hotel workers in large hotels. The research team has covered hotels in the main regions, and cities, of KSA. These main cities are Riyadh, Dammam, Abha, Jazan, Hail, Najran, and Arar. The research team decided to collect at least 1000 forms. For this reason, 1500-questionnaire forms were self-distributed with support from a specialized company in data collection. The research team distributed 30 forms in each hotel of the randomly selected 50 hotels. Responses were almost equal from the participated hotels. The research team were able to collect 1228 forms with valid responses, which represent a response rate of about 82%. This high response rate was because of the support from a specialized company in data collection.

The sample of the current study is of excellent sample size according to the suggestion of Krejcie and Morgan (1970), who suggested the sample size should be 384 for a population of 1 million. In addition, the sample size is sufficient according to the proposal of Roussel (2005), who also suggested that the number of reponses should be up to 10 times that of the items in the instrument to properly conduct factor analysis, i.e., up to 170 responses as the current study has 17 items. Data collection for the current study was in March and April 2022. Respondents of the current study were assured of confidentiality and they were informed that the data were collected for study purposes. The purpose of the study was discussed with participants before data collection. Informed consent was collected from all respondents before their participation in the study.

### 3.2. The Study Instrument

This study adopted a pre-tested research instrument for data collection. The items of the study instrument are shown in Appendix A. The transformational leadership was

examined through 7 items adopted from Carless et al. (2000). Psychological safety was tested through 7 items developed by Edmondson (2003), whereas turnover intention was assessed using three items by Liden et al. (1997). This study adopted an instrument with few items to encourage the study participants to participate. The factors were assessed through five-point Likert items, which were between 1 "completely disagree" and 5 "completely agree". This research focused on the interrelationship between these three variables and did not pay much attention to the role of participants' demographic in these relationships, which could be an opportunity for other research as will be discussed in the limitations of the study. The majority of participants were male (67%), while females were the minority (33%). The age of participants was in two categories: between 20 and 40 years old (45%) and from 41 to 60 years old (55%). There were no participants less than 20 years or above 60 years old. The majority of participants had a bachelor's degree or equivalent (62%) whereas 23% were holding a post-graduate degree (post-graduate diploma or master's degree) and 15% held a high school degree or equivalent.

The questionnaire was prepared in both English and Arabic languages. The research team adopted back translation. The translated questionnaire was double-checked by two bilingual speakers of English and Arabic for accuracy. The questionnaire form was piloted with 20 employees before full distribution to check the clarity of questions. The pilot study showed no changes in the questionnaire items. The data collected were tabulated and checked for normal distribution. The means for all responses were between 1.52 and 4.37 and standard deviations were between 0.6 and 0.954 (see Table 1), which reflect that the data are not condensed around the mean (Bryman and Cramer 2012).

**Table 1.** The descriptive statistics of the constructs (developed by authors).

| Abbr | Item | Min | Max | M | SD | Skewness | Kurtosis |
|------|------|-----|-----|---|----|----------|----------|
| | Transformational Leadership | | | | | | |
| TL1 | The leader communicates a clear and positive vision of the future | 1 | 5 | 4.23 | 0.822 | −0.964 | 0.930 |
| TL2 | The leader treats staff as individuals, supports and encourages their development | 2 | 5 | 4.23 | 0.763 | −0.524 | −0.747 |
| TL3 | The leader gives encouragement and recognition to staff | 2 | 5 | 4.32 | 0.790 | −0.933 | 0.138 |
| TL4 | The leader fosters trust, involvement and cooperation among team members | 2 | 5 | 4.15 | 0.849 | −0.764 | −0.078 |
| TL5 | The leader encourages thinking about problems in new ways and questions assumptions | 3 | 5 | 4.37 | 0.717 | −0.683 | −0.777 |
| TL6 | The leader is clear about his/her values and practices what he/she preaches | 2 | 5 | 4.16 | 0.852 | −0.776 | −0.085 |
| TL7 | The leader instills pride and respect in others and inspires me by being highly competent | 1 | 5 | 4.24 | 0.824 | −0.977 | 0.925 |
| | Turnover Intention | | | | | | |
| TU8 | I often think about leaving that career | 1 | 5 | 1.86 | 0.954 | 1.641 | 3.063 |
| TU9 | It would not take much to make me leave this career | 1 | 5 | 1.55 | 0.636 | 1.634 | 6.367 |
| TU10 | I will probably be looking for another career soon | 1 | 5 | 1.52 | 0.600 | 1.551 | 6.899 |
| | Team Psychological Safety | | | | | | |
| PS11 | If you make a mistake on this team, it is not really held against you | 3 | 5 | 4.38 | 0.718 | −0.706 | −0.760 |
| PS12 | Members of this team are able to bring up problems and tough issues | 2 | 5 | 4.22 | 0.780 | −0.514 | −0.860 |
| PS13 | People on this team never reject others for being different | 1 | 5 | 4.24 | 0.824 | −0.977 | 0.925 |
| PS14 | It is safe to take a risk on this team | 2 | 5 | 4.23 | 0.763 | −0.524 | −0.747 |
| PS15 | It is easy to ask other members of this team for help | 2 | 5 | 4.32 | 0.790 | −0.933 | 0.138 |
| PS16 | No one on this team would deliberately act in a way that undermines my effort | 2 | 5 | 4.15 | 0.849 | −0.764 | −0.078 |
| PS17 | Working with members of this team, my unique skills and talents are valued and utilized | 3 | 5 | 4.30 | 0.754 | −0.560 | −1.035 |

For the purification of the study instrument and to ensure the appropriateness of the collected data for CFA (confirmatory factor analysis) using SEM (structural equation modeling), three tests were conducted. First, the TVE (total variance explained) of the three main variables of the research was assured: "transformational leadership, turnover intention, and psychological safety" 52.31, 54.209, and 51.662, correspondingly. Second, the KMO (Kaiser–Meyer–Olkin) showed values of 0.750, 0.736, and 0.728, respectively (Appendix B). Third, the Cronbach's Alpha was adopted to ensure the reliability of the scale. As Appendix B shows, the values of alpha are 0.771, 0.780, and 0.810 for "transformational leadership, turnover intention, and psychological safety" individually. These values of Alpha mean that the scale items are highly acceptable according to Nunnally (1978). The results of TVE, KMO, and Cronbach's Alpha confirm that the collected data are appropriate for CFA. Additionally, the findings excluded the null hypothesis since the *p* value of the variables is zero. Furthermore, the two coefficients, skewness (the coefficient

of symmetry) and Kurtosis (the coefficient of flattening), were also assed to ensure normal data distribution. The findings showed that the skewness and kurtosis coefficient do not violate the assumption of normality, as suggested by Evrard et al. (2000) and Kline (2015) (see Table 1).

## 4. Key Findings

### 4.1. CFA Findings

The study adopted CFA to assess the GoF "goodness of fit" of the collected data and their fitness for the research model. These GoF metrics include: "normed chi-square" (normed χ2), "root means square error approximation" (RMSEA), Comparative Fit Index" (CFI), and "Tucker Lewis index" (TLI). Pedhazur and Pedhazur Schmelkin (1991) and Roussel (2005) argued that the normed χ2 value has to be less than 5; the SRMR has also to be less than 0.05, and the RMSEA has to be less than 0.08 and, if possible, 0.05. Moreover, Bentler and Bonett (1980) confirmed the values of NFI, the TLI, and the CFI, the threshold value of which is 0.90. The findings showed that normed χ2 = 2.70, RMSEA = 0.058, SRMR = 0.0327, CFI = 0.960, TLI = 0.968, NFI = 0.968, confirming and excellent GoF (for full information about these metrics, please see the footer of Table 2).

Convergent validity assesses whether the factors supposed to measure the same phenomenon are associated. This can be confirmed in two steps. The first step is to check that SFL for all factors are significant and exceed 0.7 (Hair et al. 2014), which is the case of the current study. The second step is to ensure that the AVE (average variance extracted) should be above 0.5 for all four factors. In the current study, the AVE was above 0.8 for all factors. The AVE values are 0.867 for transformational leadership, 0.899 for turnover intention, and 0.885 for psychological safety. These values ensured a satisfactory level convergent validity (Joreskog 1988) (see Table 2). The findings also confirmed an acceptable discriminant validity for the three factors. This was ensured through two steps. In the first step, the value of MSV (maximum shared variance) for the three factors was less than the AVE values (Fornell and Larcker 1981) (see Table 2). In the second step, the square roots of the AVEs were higher than the off-diagonal values (Table 2, in bold). Therefore, discriminant validity in this study was confirmed (Hair et al. 2014).

**Table 2.** The results of convergent and discriminant validity (developed by authors).

| Factors and Items | SFL | CR | AVE | MSV | ASV | 1 | 2 | 3 |
|---|---|---|---|---|---|---|---|---|
| 1-Transformational Leadership (Carless et al. 2000) (α = 0.771) | | 0.978 | 0.867 | 0.520 | 0.338 | **0.931** | | |
| The leader communicates a clear and positive vision of the future | 0.960 | | | | | | | |
| The leader treats staff as individuals, supports and encourages their development | 0.932 | | | | | | | |
| The leader gives encouragement and recognition to staff | 0.928 | | | | | | | |
| The leader fosters trust, involvement and cooperation among team members | 0.940 | | | | | | | |
| The leader encourages thinking about problems in new ways and questions assumptions | 0.917 | | | | | | | |
| The leader is clear about his/her values and practises what he/she preaches | 0.972 | | | | | | | |
| The leader instills pride and respect in others and inspires me by being highly competent | 0.864 | | | | | | | |
| 2-Turnover Intention (Liden et al. 1997) (α = 0.780) | | 0.964 | 0.899 | 0.667 | 0.432 | 0.189 ** | **0.948** | |
| I often think about leaving that career | 0.866 | | | | | | | |
| It would not take much to make me leave this career | 0.995 | | | | | | | |
| I will probably be looking for another career soon | 0.978 | | | | | | | |
| 3-Psychological Safety (Edmondson 2003) (α = 0.810) | | 0.982 | 0.885 | 0.667 | 0.392 | 0.591 ** | 0.776 ** | **0.940** |
| If you make a mistake on this team, it is not really held against you | 0.938 | | | | | | | |
| Members of this team are able to bring up problems and tough issues | 0.945 | | | | | | | |
| People on this team never reject others for being different | 0.966 | | | | | | | |
| It is safe to take a risk on this team | 0.918 | | | | | | | |
| It is easy to ask other members of this team for help | 0.901 | | | | | | | |
| No one on this team would deliberately act in a way that undermines my effort | 0.981 | | | | | | | |
| Working with members of this team, my unique skills and talents are valued and utilized | 0.933 | | | | | | | |

Model fit: (χ2 (18, N = 1228) = 48.6 *p* < 0.001, normed χ2 = 2.70, RMSEA = 0.058, SRMR = 0.0327, CFI = 0.960, TLI = 0.968, NFI = 0.968, PCFI = 0.643 and PNFI = 0.627); ** *p* < 0.01. SFL: Standardized Factor Loading; AVE: Average Variance Extracted; MSV: Maximum Shared Value; ASV: Average Shared Value.

### 4.2. Structural Equation Modeling Results

Two-steps verification approach was adopted in the current study. First, the study developed a theoretical model (Figure 1) based on a critical review of related literature. The study also adopted a pre-tested instrument to ensure proper data collection. Second, after checking the validity and reliability of the measures, SEM analysis was conducted to

examine the impact of transformational leadership (TL) on turnover intention (TU) through psychological safety (PS). The results of Table 3 and Figure 2 showed fit of data collected to the final structural model of the study (Figure 2). The GoF metrics for the final research model are normed $\chi^2$ = 2.6, RMSEA = 0.0229, SRMR = 0.023, GFI = 0.973, CFI = 0.961, TLI = 0.988, NFI = 0.937, PCFI = 0.710 and PNFI = 0.723 (see Table 3 and Figure 2).

The findings of the final structural model (Figure 2) showed that transformation model has a significant negative impact on turnover intention ($\beta$ = −0.39, *t*-value = 6.298, $p < 0.001$); however, it has a significant positive impact on psychological safety ($\beta$ = +0.72, *t*-value = 11.315, $p < 0.001$). Additionally, psychological safety has a significant negative impact on turnover intention ($\beta$ = −0.42, *t*-value = 3.674, $p < 0.001$). Furthermore, the robustness of the final model is legitimized by value $R^2$ for turnover intention ($R^2$ = 0.759) (Table 3). This means that the adoption of psychological safety and transformational leadership explains 76% of turnover intention variance.

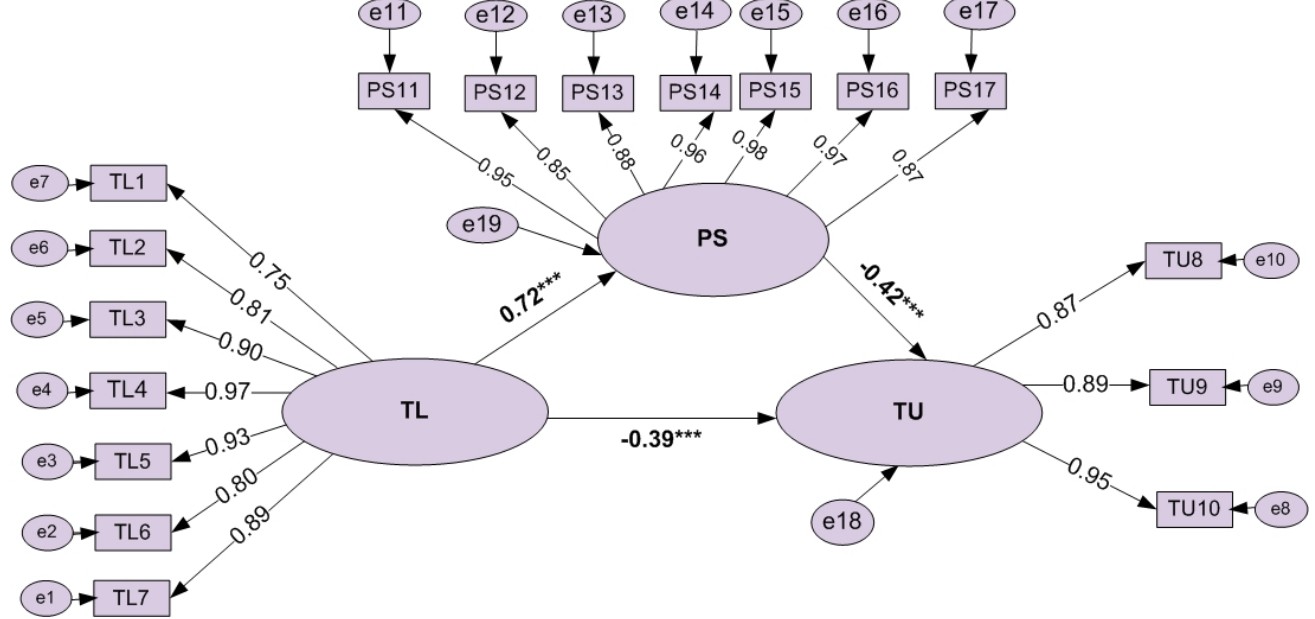

**Figure 2.** The structural model (developed by authors). *** $p < 0.001$.

**Table 3.** The result of structural model (developed by authors).

| Result of the Structural Model | β | C-R *t*-Value | R² | Hypotheses |
|---|---|---|---|---|
| H1—Transformational leadership ⟶ turnover intention | −0.39 *** | 6.298 | | Supported |
| H2—Transformational leadership ⟶ psychological safety | 0.72 *** | 11.315 | | Supported |
| H3—Psychological safety ⟶ turnover intention | −0.42 *** | 3.674 | | Supported |
| Turnover Intention | | | 0.759 | |

Model fit: ($\chi^2$ (25, N = 1228) = 65 $p < 0.001$, normed $\chi^2$ = 2.6, RMSEA = 0.0229, SRMR = 0.023, GFI = 0.973, CFI = 0.961, TLI = 0.988, NFI = 0.937, PCFI = 0.710 and PNFI = 0.723); *** $p < 0.001$.

For examining the mediating effect of psychological safety in the link between transformational leadership and turnover intention, the current study used the methodology proposed by Baron and Kenny (1986). First, the relationship between transformational leadership and turnover intention was assessed to be significant to show a possibility of mediation. This was confirmed as transformational leadership has a significant negative effect on turnover intention ($\beta$ = −0.39, *t*-value = 6.298, $p < 0.001$). Second, the impact of transformational leadership on psychological safety was assessed to be statistically significant as the findings confirmed ($\beta$ = +0.72, *t*-value = 11.315, $p < 0.001$). Third, the relationship between the mediating variable, i.e., psychological safety and turnover intention, has to be significant, which was in the current study ($\beta$ = −0.42, *t*-value = 3.674,

$p < 0.001$). The next step is to check the type of mediation, whether partial or full mediation. The finding in Table 4 shows that the impact of transformational leadership on turnover intention is no longer significant after the provision of psychological safety as a mediator variable ($\beta = -0.318$, $p = 0.059 > 0.05$). Thus, it was found that the mediation effect of the psychological safety is full or perfect mediation between transformational leadership and turnover intention.

**Table 4.** The results of mediation effect.

| Parameter | Estimate | Lower | Upper | $p$ | Mediation |
|---|---|---|---|---|---|
| H4—Transformational leadership ⟶ turnover intention ⟶ psychological safety | 0.318 | 0.145 | 0.411 | 0.059 | 0.059 > 0.05 Perfect Mediation |

## 5. Discussion and Implications

Workers of any organization are the most valuable asset. This is especially true for the hotel industry, which depends on their workers for the provision of quality services. However, hotels are suffering from high workers' turnover rate, mainly due to poor working environmental conditions compared to other industries (Sobaih 2011; Haldorai et al. 2019). Hence, the industry has received a special attention for scholars to understand the antecedents of this turnover as well as the intention to turnover. This matter has become more important recently due to the negative impacts of COVID-19 pandemic on workers, including their turnover intention. Thus, this study has an essential area of focus where leadership style, particularly a transformational one, should lead in such a way as to enhance followers' psychological safety and decrease their turnover intention. The COVID-19 pandemic has had hit the economy and workers in many industries, and the hotel industry has been among the most hit businesses (Sobaih et al. 2021). This study attempts to understand the high turnover rate in the hotel industry amid the COVID-19 pandemic. Little research has been conducted to examine the role of psychological safety as a mediator between transformational leadership and turnover intention by drawing on the SET framework (Blau 1968). Thus, this study adds to the body of literature by integrating theories of transformational leadership with SET to test the interrelationship between transformational leadership, psychological safety, and turnover intention.

Several significant outcomes emerged from this study. More specifically, the study proposed four hypotheses. The first hypothesis suggested a negative impact of transformational leadership on workers' turnover intention. As anticipated, the results supported this hypothesis and previous literature pre-COVID-19 (e.g., Chen and Wu 2017; Zou et al. 2015; Tang et al. 2015; Sobaih et al. 2022). The pandemic of COVID-19 has created an extensive sense of job insecurity among hotel employees, which has had a significant impact on their intention to leave the hotel or changing their career (Aliedan et al. 2022). Transformational leadership has a strong connection with followers' attitude and behavior through their motivation, support, inspiration, training, care, and values, which contribute to connections with the members of the organization and affect their attitude and behavior (Sobaih et al. 2022). Likewise, the study findings support the second research hypothesis that transformational leadership has a positive impact on workers' psychological safety. The result was expected, because followers have the confidence, feel safe, and the trust without fears in their leaders, which translated into psychological safety. Third, the findings of current study support the third hypothesis that suggested psychological safety has a negative impact on employees' turnover intention. Psychological safety has the ability to limit the intention to leave among hotel workers.

One of the main objectives and questions in this research was to investigate the mediating effect of psychological safety on the link between transformational leadership and employee turnover intention in the hotel industry. The findings, interestingly, showed a perfect mediating effect of psychological safety in the relationship between transformational leadership and employee turnover intention. Therefore, psychological safety can

unlock the relationship between transformational leadership and turnover intention. The findings confirmed that the link between transformation leadership and turnover intention could be controlled by the state of psychological safety among hotel workers. Psychological safety has the ability to decrease negative outcomes for employees, even in the absence of transformational leadership. Drawing on SET theory, transformational leadership stimulates psychological safety among their team members, and consequently followers engage in positive attitudes and intention to stay in the organization (Sobaih et al. 2022). It is interesting that psychological safety can unlock the "black box" between leadership style and job outcomes, e.g., turnover intention (Sobaih et al. 2019). The existence of psychological safety ensures lower turnover intention, even during crises, e.g., the COVID-19 pandemic.

The findings of the current study have several theoretical implications. First, the current study is the first of its kind to test the mediating effect of psychological safety in the relationship between transformational leadership and turnover intention amid COVID-19, especially in a growing economy, such as the hotel industry in KSA. Most research studies focused on direct relationships and mainly in the developed country context. Second, the research confirmed a strong linear connection and a direct relationship of transformational leadership with turnover intention. This means that transformational leaders have the ability to lower the turnover intention among their workers though transformational leadership practices. Third, the results of this research confirmed a full mediating effect of psychological safety in the link between transformational leadership and turnover intention. This highlights the crucial role of psychological safety in affecting, albeit controlling, the above relationship and motivates scholars to undertake upcoming research on the antecedent's psychological safety to minimize the turnover intention among hotel workers. Finally, this study examined the aforementioned relationships in the context hotel workers amid the pandemic of COVID-19, and thus makes an important contribution to the theoretical hospitality and tourism literature, especially in relation to the crucial role of psychological safety in turnover intention and appropriate working environment.

The current study results contribute significantly to practical development by providing some suggestions and guidance to hotel managers on making proper working environments for their workers. This could be translated into creating an environment to stimulate psychological safety and reduce the perceptions of turnover intention among workers. Hotel managers should focus on transformational leadership practices that turn to stimulate and retain employees in the organization. Leaders in hotels should make psychological safety an explicit priority. They have to effectively create the conditions of psychological safety for their teams. This can be done by creating an environment where workers can speak up and ask questions with no fear of being punished. In addition, leaders should acknowledge ideas of their workers and use them to create a positive experience. They need to establish norms around how failure is handled. Such practices create a safe environment for workers at their hotels. It is important that leaders adopt practices of transformational leadership such as individualized consideration, fair treatment, intellectual stimulation, and inspirational motivation, to raise workers' level of psychological safety as well as reduce perceptions of turnover intention. Moreover, the findings also contributed to hotel managerial practices by revealing that transformational leadership has an effective pathway to reduce employees' intention of turnover. Amid the pandemic of COVID-19 pandemic, it has been detected that hotel workers feel unsafe and job insecure, hence this research demonstrated the importance of transformational leadership practices and psychological safety in order for hotels to retain their valuable asset: workers. In line with this experience, hotel executive management should invest in their transformational leader, who has the ability to create supportive working environments and enhance the psychological safety of their team.

## 6. Conclusions and Limitation

Staff turnover in hospitality industry is almost double the average rate for other industries and this rate has increased amid the pandemic of COVID-19. Thus, although

the topic of workers' turnover intention in the hotel context is not new, it has become a hot area for research amid the COVID-19 pandemic. The current research results have meaningfully contributed to the implementation of leadership theories as a strategy to minimize turnover intention among workers in the hotel industry. More specifically, this study examined the mediation effect of psychological safety in the link between transformational leadership and turnover intention in the hotel context. The study findings demonstrate that transformational leadership predicts turnover intention negatively. Additionally, psychological safety has a negative effect on turnover intention. On the other side, psychological safety has a perfect mediation between transformational leadership and turnover intention. This means that the existence of psychological safety ensures the occurrence of turnover intention. This also means that psychological safety has the ability to control the link between transformational leadership and turnover intention. Psychological safety can change the influence of transformational leadership on turnover intention. Hence, it has to have higher attention from transformation leaders to ensure the retention of their workers. Transformational leadership styles have the ability to support, inspire, care, treat fairly, and engage all workers, which in turn create a psychological safety environment and ultimately decreases employees' intention to leave.

Like other social science research, this research encountered some limitations; however, the limitations can open the door for upcoming research opportunities. First, the data collection has conducted in large hotels only located in a specific country, exactly, i.e., KSA. Hence, attention is required when generalizing the results to the contexts of other countries. In particular, this study did not pay sufficient attention to cultural differences; thus, future research could investigate the research model across other cultural contexts. Second, the study did not examine variables such as the impact of the demographics of participants, for example gender, experience, and age in understanding the role of psychological safety in the link between transformational leadership and intention of turnover. Therefore, the upcoming studies could examine the moderating effect of these demographics. Third, the study objective was to examine the mediating effect of psychological safety in the link between transformational leadership and turnover intention. Hence, future study may try to investigate a different style of leadership or to incorporate other factors as determinants of turnover intention in the workplace. For instance, job satisfaction; trust in organization; perceived organization support; and job engagement.

**Author Contributions:** Conceptualization, A.E.E.S., H.G. and A.E.A.E.; methodology, A.E.E.S., H.G., and A.E.A.E.; software A.E.E.S. and H.G.; validation, A.E.E.S. and A.E.A.E.; formal analysis, A.E.E.S. and H.G.; investigation, A.E.E.S., H.G. and A.E.A.E.; resources, A.E.E.S. and H.G.; data curation, A.E.E.S., H.G. and A.E.A.E.; writing—original draft preparation, A.E.E.S., H.G., and A.E.A.E.; writing—review and editing, A.E.E.S. and A.E.A.E.; visualization, A.E.E.S. and A.E.A.E.; supervision, A.E.E.S., H.G. and A.E.A.E.; project administration, A.E.E.S.; funding acquisition, A.E.E.S., H.G. and A.E.A.E. All authors have read and agreed to the published version of the manuscript.

**Funding:** This work was supported by the Deanship of Scientific Research, Vice Presidency for Graduate Studies and Scientific Research, King Faisal University, Saudi Arabia [GRANT944].

**Institutional Review Board Statement:** The study was conducted according to the guidelines of the Declaration of Helsinki and approved by the Deanship of Scientific Research Ethical Committee, King Faisal University (project number: GRANT944, date of approval: 2 March 2022).

**Informed Consent Statement:** Informed consent was obtained from all subjects involved in the study.

**Data Availability Statement:** Data is available upon request from researchers who meet the eligibility criteria. Kindly contact the first author privately through the e-mail.

**Conflicts of Interest:** The authors declare no conflict of interest.

## Appendix A

**Table A1.** The measurement scales.

| Abbr | Scales and Items | Authors |
|------|------------------|---------|
| | Transformational Leadership | |
| TL1 | The leader communicates a clear and positive vision of the future | |
| TL2 | The leader treats staff as individuals, supports and encourages their development | |
| TL3 | The leader gives encouragement and recognition to staff | |
| TL4 | The leader fosters trust, involvement and cooperation among team members | Carless et al. (2000) |
| TL5 | The leader encourages thinking about problems in new ways and questions assumptions | |
| TL6 | The leader is clear about his/her values and practises what he/she preaches | |
| TL7 | The leader instills pride and respect in others and inspires me by being highly competent | |
| | Turnover Intention | |
| TU8 | I often think about leaving that career | |
| TU9 | It would not take much to make me leave this career | Liden et al. (1997) |
| TU10 | I will probably be looking for another career soon | |
| | Psychological Safety | |
| PS11 | If you make a mistake on this team, it is not really held against you | |
| PS12 | Members of this team are able to bring up problems and tough issues | |
| PS13 | People on this team never reject others for being different | |
| PS14 | It is safe to take a risk on this team | Edmondson (2003) |
| PS15 | It is easy to ask other members of this team for help | |
| PS16 | No one on this team would deliberately act in a way that undermines my effort | |
| PS17 | Working with members of this team, my unique skills and talents are valued and utilized | |

## Appendix B

**Table A2.** KMO, Total Variance Explained and Cronbach Alpha.

| Measured Variable | KMO | TVE | $\alpha$ |
|-------------------|-----|-----|----------|
| Transformational Leadership = TL | 0.750 | 52.318 | 0.771 |
| Turnover Intention = TU | 0.736 | 54.209 | 0.780 |
| psychological Safety = PS | 0.728 | 51.662 | 0.810 |

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
