# Peer review of "Do You Feel Safe Here? The Role of Psychological Safety in the Relationship between Transformational Leadership and Turnover Intention Amid COVID-19 Pandemic"

_jrfm, doi:10.3390/jrfm15080340_

Round 1
Reviewer 1 Report
The topic is exciting and well articulated. However, several things can be considered to improve the paper.
1. The paper title can be shortened.
2. The B, t and p values should be removed from the abstract.
3. Theoretical perspective is missing in the manuscript. There should be a discussion on the core theory used in this study.
4. The research problem presented in the introduction is not very convincing and solid. The authors need to justify why these findings can be generalized for a post covid situations.
5. The implications of the findings for scholars and practitioners are rather general and need to be very specific to the research setting.
Author Response
Dear Reviewer,
Thank you for giving us the opportunity to resubmit a revised draft of our manuscript entitled “Do You Feel Safe Here? The Role of Psychological Safety in the Relationship between Transformational Leadership and Turnover Intention amid COVID-19 Pandemic” for publication consideration in the special issue “‘Business Performance’ at the journal “Journal of Risk and Financial Management“.
We appreciate the time and effort that you have dedicated to provide us with your valuable feedback on our manuscript. We are grateful for your insightful comments on our paper. We have been able to incorporate changes to reflect the suggestions provided in our revised manuscript. We have used track changes to easily check changes. Please also note that we have incorporated the feedback of the other reviewers in this revised version.
Thank you once again for your insightful input and constructive feedback.
Attached is a point-by-point response to your comments and concerns.
Best regards
The authors

Reviewer 2 Report
The paper is well written and the research well presented. I only would suggest the author to better describe the characteristics of the sample units, i.e.: on how many hotels was the survey performed? Are there employees belonging to the same hotel? If yes, how this could affect the results? On this last point for example the authors could mention the possibility (as future research avenue) of considering a multi-level modeling approach.
Individual covariates of employees are not taken into account in model estimation. Could this inclusion affect the results? This limit is highlighted in the conclusion section, but the authors could better explain why they decided of not considering demographic and other individual characteristics in their analysis.
There are some typos, so I recommend the authors to check the paper. Below some examples:
- Line 141, I would suggest adding a comma after Ajzen and Fishben (1980) theory,
- Line 146, Add a comma after According to Khan (2015),
- Line 210. On he other side ("On" should be capital)
- Line 216 "Amid" should not be capital
- Line 217 "greater stress increases"
- Line 430 "CVOID-19"
Author Response

(The authors gave the same response as above.)
